# German Porphyria Registry (PoReGer)–Background and Setup

**DOI:** 10.3390/healthcare12010111

**Published:** 2024-01-03

**Authors:** Lea Gerischer, Mona Mainert, Nils Wohmann, Ilja Kubisch, Ulrich Stölzel, Thomas Stauch, Sabine von Wegerer, Fabian Braun, Christina Weiler-Normann, Sabine Blaschke, Jorge Frank, Rajan Somasundaram, Eva Diehl-Wiesenecker

**Affiliations:** 1Department of Neurology, Charité–Universitätsmedizin Berlin, Corporate Member of Freie Universität Berlin and Humboldt-Universität zu Berlin, 10117 Berlin, Germany; 2Neuroscience Clinical Research Center, Charité–Universitätsmedizin Berlin, Corporate Member of Freie Universität Berlin and Humboldt-Universität zu Berlin, 10117 Berlin, Germany; 3Department of Emergency Medicine and Porphyria Clinic, Charité–Universitätsmedizin Berlin, Corporate Member of Freie Universität Berlin and Humboldt-Universität zu Berlin, 12203 Berlin, Germany; 4Department of Internal Medicine II, Saxony Porphyria Center, Klinikum Chemnitz, gGmbH, 09116 Chemnitz, Germany; 5Porphyria Laboratory IPNET, MVZ Labor PD Dr. Med. Volkmann GbR, 76131 Karlsruhe, Germany; 6Patient Organization, Berliner Leberring e.V., 12203 Berlin, Germany; 7Martin Zeitz Center for Rare Diseases, University Medical Center Hamburg-Eppendorf, 20246 Hamburg, Germany; 8III. Department of Medicine, University Medical Center Hamburg-Eppendorf, 20246 Hamburg, Germany; 9Emergency Department, University Medical Center Göttingen, 37075 Göttingen, Germany; 10Department of Dermatology, Venereology and Allergology, University Medical Center Göttingen, 37075 Göttingen, Germany

**Keywords:** porphyria, acute porphyrias, rare disease registry, epidemiology

## Abstract

Porphyrias, as most rare diseases, are characterized by complexity and scarcity of knowledge. A national registry in one of the largest European populations that prospectively collects longitudinal clinical and laboratory data are an important and effective tool to close this gap. The German Porphyria Registry (PoReGer) was founded by four centers with longstanding expertise in the field of porphyrias and rare diseases (Charité–Universitätsmedizin Berlin, Porphyria Center Saxony Chemnitz, University Medical Center Hamburg-Eppendorf, University Medical Center Göttingen) and the German reference laboratory for porphyria, and is supported by the largest German porphyria patient organization. A specified data matrix for three subgroups (acute, chronic blistering cutaneous, acute non-blistering cutaneous) includes data on demographics, specific porphyria-related symptoms, clinical course, general medical history, necessary follow-up assessments (including laboratory and imaging results), symptomatic and disease-modifying therapies, and side-effects. Additionally, the registry includes patient-reported outcome measures on quality of life, depression, and fatigue. PoReGer aims to broaden and deepen the understanding on all porphyria-related subjects. We expect these data to significantly improve the management and care of porphyria patients. Additionally, the data can be used for educational purposes to increase awareness, for the planning of healthcare services, and for machine learning algorithms for early detection of porphyrias.

## 1. Introduction

Porphyrias are a heterogeneous group of rare metabolic diseases that result from enzyme deficiencies in the heme biosynthesis pathway. Rarity and variate symptoms may result in misdiagnosis or delayed diagnosis, and correct diagnosis and treatment need highly specialized and interdisciplinary care [1,2,3]. Despite a longstanding history of porphyria specialists in Germany (e.g., Doss-Porphyria [4,5,6]) and recent international efforts of standardizing management recommendations [7], prospective data are scarce and important questions, e.g., on predicting clinical courses and therapy response, are largely unresolved. The aim of our network is to improve the diagnosis and interdisciplinary care for patients with porphyria in Germany. One important step is the foundation of the German Porphyria Registry (PoReGer).

With a prevalence of diagnosed porphyrias between 1-10/1,000,000, most porphyria subtypes are rare or ultra-rare diseases [8,9]. Most forms are inherited, except the acquired form of porphyria cutanea tarda (PCT) [10]. In total, there are eight main different forms of porphyria. Each form of porphyria is defined by an enzymatic abnormality (deficiency or gain of function) in one of the eight successive enzymes of heme metabolism resulting in an accumulation of mostly toxic heme precursors or porphyrins. Historically, porphyrias have been categorized as hepatic or erythropoietic [1]. Recently, a clinically more relevant categorization into acute (neurovisceral) porphyrias, chronic blistering cutaneous porphyrias and acute non-blistering cutaneous porphyrias has been proposed [7,9,11,12,13].

The acute porphyrias (AP) comprise four forms—acute intermittent porphyria (AIP, OMIM #176000, autosomal-dominant), variegate porphyria (VP, OMIM #176200, autosomal-dominant) hereditary coproporphyria (HCP, OMIM #121300, autosomal-dominant), and ultra-rare 5-aminolevulinic-acid-dehydratase-deficient porphyria (ALADP/Doss-Porphyria, OMIM #612740, autosomal recessive). Patients with APs experience acute, possibly life-threatening attacks with neuro-psychiatric, abdominal, and cardiovascular symptoms, characterized by significant accumulation of porphyrin precursors aminolevulinic-acid (ALA) and porphobilinogen (PBG). HCP and VP can additionally cause cutaneous symptoms that are clinically indistinguishable from PCT [9,12,14]. Acute attacks usually manifest after puberty and can be induced by several porphyrinogenic drugs, fasting, alcohol consumption, smoking, all forms of stress (surgery, infection), sexual hormones, and xenobiotics [7,9,11,14]. More than 75% of patients with acute neurovisceral attacks need to be hospitalized [3]. Urgent diagnosis and therapeutic measures are required to prevent severe neurological complications [3,14].

The cutaneous porphyrias can be further categorized into chronic blistering and acute non-blistering forms. The chronic blistering forms comprise porphyria cutanea tarda (PCT, type I ‘sporadic’, OMIM #176090 and type II ‘familial’, OMIM #176100, autosomal-dominant), hepatoerythropoietic porphyria (HEP, OMIM #176100, autosomal-recessive), and ultra-rare congenital erythropoietic porphyria (CEP/Gunther Disease, OMIM # 263700, autosomal recessive). Erythropoietic protoporphyria (EPP, OMIM #177000, autosomal-recessive) and X-linked protoporphyria (XLEPP, OMIM #300752, X-linked) represent the acute non-blistering cutaneous porphyrias. XLEPP is considered a subtype of EPP and accounts for up to 10% of cases [15]. (There is one further ultra-rare form, described in single cases, the (erythropoietic) Harderoporphyria (OMIM # 618892, autosomal-recessive), which is omitted in common classifications. Rarer genetic protoporphyrias (e.g., EPP2, OMIM # 618015) are subsumed under the umbrella term EPP/XLEPP, too.) The main symptom of all cutaneous porphyrias is skin photosensitivity. Some forms can cause relevant anemia and varying degrees of liver damage [9,11,12,16,17]. In most cases, EPP/XLEPP, HEP, and CEP manifest in infancy or early childhood [12,15,16,17], whereas PCT commonly manifests in the 5th decade of life [10]. In contrast to the other forms, PCT is mostly an acquired disease due to iron overload and other susceptibility factors (such as alcohol abuse, smoking, hemochromatosis due to pathogenetic HFE gen variants, hepatitis C virus or HIV infection) but further gene mutations may also play a role [10]. EPP/XLEPP, as well as PCT, can lead to liver damage including cirrhosis. PCT patients, in addition, have an independently elevated risk for hepatocellular carcinoma (HCC). In severe cases of EPP, bone marrow and/or liver transplantation have to be performed [10,15].

Diagnosis of all forms of porphyrias can effectively be made via the identification of specific biochemical patterns of elevated porphyrins and porphyrin precursors in urine, feces, and blood. Furthermore, a plasma fluorescence scan can support the biochemical diagnosis [9,18].

Awareness of these rare diseases is low and leads to delays in diagnosis. Patients suffer from chronic complications, significant disease burden, and decreased quality of life. In APs, fast and accurate diagnosis is paramount to initiate proper treatment of potentially life-threatening attacks. Timely diagnosis and adequate management in all forms helps to prevent liver damage and/or HCC. Moreover, in Aps, givosiran is now available as disease modifying therapy, and EPP/XLEPP can be ameliorated with afamelanotide [9,10,11,15,19,20].

The aim of this registry is to gather detailed multicentric prospective and retrospective data on initial symptoms, diagnostics, therapies—especially in the light of evolving new drugs—as well as adverse effects, long-term outcomes, predicting and preventing factors, comorbidities, and quality of life. We expect these data to significantly improve the management and care of porphyria patients. In future, this extensive dataset could additionally be used for machine learning algorithms for early detection of porphyrias. This manuscript illustrates the setup, structure, and governance of the German Porphyria Registry.

## 2. Methods

### 2.1. Founding Centers of the PoReGer-Registry

Our network consists of a core of four founding centers, all with longstanding clinical expertise in the field of porphyrias: Charité–Universitätsmedizin Berlin (hepatologic, neurologic, dermatologic and emergency care specialists), Porphyria Center Saxony, Chemnitz (hepatologic and gastroenterologic specialists), University Medical Center Hamburg-Eppendorf, Hamburg (hepatologic and nephrologic specialists), and University Medical Center Göttingen (dermatologic and emergency care specialists), as well as the German reference laboratory for porphyria (International Porphyria Network (IPNET) member) in Karlsruhe. Further specialists from additional fields are regularly engaged in clinical case discussions and therapy decisions (e.g., gynecologists and pain specialists). The network has been working together for several years now and we have established regular online case-conferences between the centers as well as research collaborations. An important addition is the regular involvement of a German porphyria patient organization (Berliner Leberring e.V., Berlin, Germany). Representatives of this patient organization were also involved in the development of the data matrix for the registry.

### 2.2. Set-Up and Structure of the Registry

The PoReGer-registry has been set up using REDCap^®^ (Research Electronic Data Capture, Version 13.7.27), a secure web application designed to collect and manage scientific data worldwide. Data can be collected longitudinally and managed and updated in a multicenter setting. REDCap^®^ offers a survey option, enabling patients to answer questionnaires on their own electronic devices, and the data are then sent directly into the database.

We decided to categorize the porphyrias within the registry into the three above-mentioned categories: acute (AIP, VP, HCP, ALADP), chronic blistering cutaneous (PCT, CEP, HEP), and acute non-blistering cutaneous (EPP/XLEPP) porphyrias. For each subgroup, a specified data matrix was used that includes demographic data, data on specific porphyria-related symptoms and clinical course, general medical history, necessary follow-up assessments (including laboratory and imaging results), and data on symptomatic and disease-modifying therapies and side-effects (see Table 1). Additionally, the registry includes questionnaires for quality of life and common comorbidities like fatigue and depression. Each data matrix has been developed by interdisciplinary specialists available through our network and has been reviewed by all four participating centers and a representative of the patient organization. Forms and fields can be modified in the future; e.g., by adding new questionnaires for a specific project.

### 2.3. Study Population—Inclusion and Exclusion Criteria

Every patient with a confirmed porphyria subtype can be included in the registry. Inclusion and exclusion criteria are listed in Table 2. Diagnosis made externally—not from participating centers—must be validated before inclusion. Patients will mostly be recruited via the local outpatient clinics and emergency departments of the specialized participating centers.

### 2.4. Ethical Considerations and Data Protection

The registry was approved by the ethics committee of the Charité–Universitätsmedizin Berlin (EA 1/213/22) and by the local institutional ethics committees of two other participating centers. Patient inclusion and data collection will be in accordance with the Declaration of Helsinki in its currently applicable form. The participation in the registry is invariably voluntarily. Patients have to provide written informed consent prior to data collection/inclusion. Consent can be withdrawn at any time and without giving a reason.

After inclusion, each patient is assigned a unique identification number as part of the pseudonymization process and only pseudonymized data are entered into REDCap^®^. Each participating center can only access the identification list of their own patients. Additionally, the data protection concept of the software includes a user-separated and password protected operation.

During routine clinical care samples are being sent to and analyzed at the German reference laboratory for porphyria in Karlsruhe. The results will be included based on the consent given by the patients to participate in the registry.

### 2.5. Government of PoReGer

The four participating centers form the consortium of the registry (Figure 1). The aim of this consortium is to operate and further develop the registry. The steering group acts on behalf of the consortium as head of the registry and managing body. Each member (=each center) has one vote. Furthermore, at the Charité Berlin, a project board of at least two persons not entitled to vote are appointed to administer the registry. Regular meetings are scheduled by the project board with at least one (entitled) attending participant from every collaborating center and one (non-entitled) member of the project board. At these meetings the inclusion progress, potential changes and developments of the registry, and specific scientific analyses will be discussed and consented to. Every collaborating center has the right to deny usage of its contributed data for a specific analysis. However, the overall aim is to enable high-quality multicentric analyses with registry data from as many patients as possible.

## 3. Discussion

Porphyrias are a group of largely hereditary diseases of the heme biosynthesis pathway. As in most rare diseases, there is a lack of objective information on prevalence, disease course, complications, response and non-response to therapeutic strategies, disease burden, and impact on quality of life. Patient registries are a key tool for gathering this scarce knowledge to improve the understanding of the disease and treatment available to patients. In fact, in 2014, the European Union Committee of Experts on Rare Diseases (EUCERD) recommended the creation of registries for rare diseases to improve the management and care of patients with rare diseases. Beyond that, such registries can serve as a communication and networking tool for treating specialists and to a lesser degree for affected patients [21].

In setting up a national clinical registry for porphyrias, we aim to broaden and deepen the understanding on all porphyria-related subjects. The gathered information can additionally be used for educational purposes, for planning of healthcare services for these diseases, and possibly for the development of diagnostic algorithms including utilization of artificial intelligence. By increasing awareness for porphyrias the challenges of underdiagnosis, misdiagnosis, and diagnostic latency can be addressed.

### 3.1. Founding Centers, Set-Up, Patient Population

The founding centers have a longstanding expertise in the field of porphyrias and/or rare diseases. Furthermore, the authors recently conducted a pilot study on the concept of early detection of porphyrias in the emergency department [20]. Moreover, the German patient organization (Berliner Leberring e.V.) has regularly advised the consortium on data and questionnaires that focus on the patient perspectives and on patient empowerment. PoReGer is set up to collect prospective and longitudinal data and is focused on clinical data and measures of quality of life. Additionally, the set of common data elements for rare diseases registration from the European Platform on Rare Disease Registration were included (“https://eu-rd-platform.jrc.ec.europa.eu/set-of-common-data-elements_en” (accessed on 15 November 2023)). Additionally, recently published key terms and definitions for acute porphyrias were integrated in the registry matrix [7].

In 2001, the European Porphyria Initiative (EPI), a collaborative network of porphyria centers, was formed. In 2007, the network evolved into the European Porphyria Network (EPNET), and in 2023 into the International Porphyria Network (IPNET). Participating centers are required to adhere to agreed quality criteria [22]. IPNET has integrated clinical centers but remains focused on laboratory issues. Thus, PoReGer adds to these data via the integration and focus on clinical data. The collection of longitudinal and prospective data are comparable to several other longstanding European registries, mainly in Scandinavian countries, such as the Norwegian Porphyria Center (NAPOS) Registry collecting clinical data on all porphyrias [23,24], the Swedish national registry on porphyria gene carriers [25], a Swedish–Danish cohort study on mortality risks of porphyrias [26] and the evaluation of all known Finnish patients with acute porphyrias [27]. PoReGer is based on this knowledge and is designed to structurally evaluate different issues (demographics, genetics, clinical picture, necessity and extent of follow-up assessments, secondary conditions and long-term consequences, availability and effectiveness of symptomatic and disease-modifying therapies and their side-effects) with a yearly follow-up. Furthermore, to our knowledge, it is the first porphyria registry that includes longitudinal data on quality of life and important comorbidities like fatigue and depression.

Moreover, Germany is one of the largest countries in Europe with potential country-specific particularities, e.g., regarding inheritance or clinical courses in all forms of porphyria. A total of approximately 200–300 patients with porphyria are regularly seen and treated at the founding centers (Berlin, Chemnitz, Göttingen, Hamburg, Germany). With an estimated number of 1500 patients diagnosed with porphyria in Germany (unpublished data from the German reference laboratory for porphyria in Karlsruhe), we estimate that initially at least 200 patients will be recorded in the registry by the end of 2024. In the near future, we anticipate up to 300 more patients (including newly diagnosed cases).

### 3.2. Strengths and Limitations

The registry is set up according to all applicable ethical standards. Data protection is ensured within REDCap^®^, a secure web application, with audit log and a pseudonymization process enabling only designated study team members to access identifying data from their own center. Furthermore, data access is limited for each study center to its own data. Thus, each study center has sovereignty over their data. Collaboration on a research topic, collective data analyses, and subsequent publications are only possible with the consent of each particular center, guaranteed via legal contracts and the set-up of a steering committee.

The main limitation is the hitherto low number of participating centers, which makes it difficult to reach all porphyria patients in Germany. This is especially as diagnosis is not always made by a (tertiary care) center but sometimes rather by smaller clinics or private practices. However, many patients with rare diseases are at some point referred to specialist centers.

### 3.3. Future Plans

The first patient was included into the registry in August 2023 at the Charité. The three participating centers, Göttingen, Chemnitz, and Hamburg, will sign their collaboration contracts in the beginning of 2024. The first meeting of the steering board will then take place in spring 2024. Once the registry is working at the four founding centers, we plan to include more centers in Germany. To ensure data quality, future stability, and commitment and, since this kind of extensive data collection with yearly follow-ups is time consuming, applications for national or European funding will follow. As an important further step towards the integration of the registry in Europe, we firstly envisage to expand the network within our specialist center network in the German-speaking part of Europe (Germany, Austria, and Switzerland—“DACH”) and secondly collaborate with further European and non-European countries.

## 4. Conclusions

The German Porphyria Registry is a fully set-up and functioning national rare disease registry. The data will improve the understanding of these complex conditions and available therapies. Furthermore, it will enable measures to offer tailored specialist services for affected patients addressing their special needs in terms of quality of life.

## Figures and Tables

**Figure 1 healthcare-12-00111-f001:**
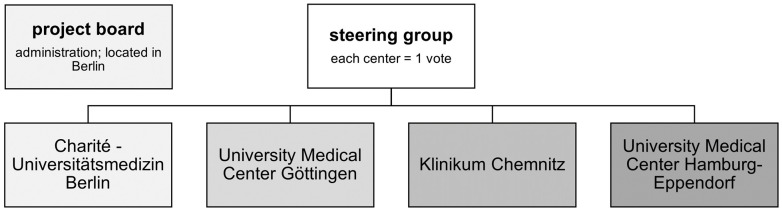
Government and structure of the consortium.

**Table 1 healthcare-12-00111-t001:** Overview of data entered into the registry.

**Demographics**
Age, sex, body mass index
social status, education, first digit of postal code, degree of disability
**Diagnosis**
Date/year of first manifestation of symptoms and date/year of diagnosis
Diagnosis made by which specialty
Gene mutation, family history
**Disease manifestation and clinical course**
Symptoms at first manifestation and in the further course
Number of acute attacks (APs)
Chronic symptoms
Number and frequency of medical treatment and hospitalizations
Trigger factors and measures to prevent them
Influence of menstrual cycle/female hormones on symptoms (APs)
**Investigations**
Physical examination
Laboratory results (general and porphyria specific)
Imaging and liver investigations (ultrasound, magnetic resonance imaging (MRI), elastography, liver biopsy)
Dermatological examination and skin type (Fitzpatrick-Scale)
Neurological examination
Results of neurological investigations, if necessary (cranial computed tomography/MRI, nerve conduction studies)
Gynecological examination and history
**Therapies for porphyria**
Symptomatic and disease modifying treatments/interventions
Side effects
**Medical history**
comorbidities
Medications for comorbidities
History of surgical interventions (e.g., appendectomy, gall bladder removal, etc.)
History of alcohol consumption, smoking status, illicit drugs
**Questionnaires**
Quality of life (EQ-5D-5L)
Depression (PHQ-9)
Fatigue (FAS)

Abbreviations: APs: acute porphyrias; EQ-5D-5L: European Quality of Life Questionnaire–5 Dimensions–5 Levels; FAS: fatigue assessment scale; PHQ-9: patient health questionnaire–short version with 9 items.

**Table 2 healthcare-12-00111-t002:** Inclusion and exclusion criteria.

**Inclusion Criteria**
○ Written informed consent
○ Confirmed diagnosis of one form of porphyria (AIP, VP, HCP, ALADP, EPP/XLEPP, CEP, HEP, PCT)
○ Ae ≥ 18 years
**Exclusion Criteria**
○ Withdrawal of study consent by the patient

Abbreviations: AIP: acute intermittent porphyria, ALADP: 5-aminolevulinic-acid-dehydratase-deficient porphyria, CEP: congenital erythropoietic porphyria, EPP: erythropoietic protoporphyria HCP: hereditary coproporphyria, HEP: hepatoerythropoietic porphyria, PCT: porphyria cutanea tarda, VP: variegate porphyria, XLEPP: X-linked erythropoietic protoporphyria.

## Data Availability

Data are contained within the article.

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
