# Peer review of "German Porphyria Registry (PoReGer)–Background and Setup"

_healthcare, 2024, doi:10.3390/healthcare12010111_

Round 1
Reviewer 1 Report
Comments and Suggestions for Authors
This is a terrific effort and a well-written article. Thank you for sharing this plan, and I wish the group continued success with studying porphyria among this patient cohort. I suggest the following minor edits:
1. For line 53, you say that the prevalence is 1-10/1,000,000 for most subtypes; however, this is just for people diagnosed with porphyria. Each type is likely underdiagnosed, so please edit the sentence to clarify. For instance, you could say the following: “With a prevalence of diagnoses between 1-10/1,000,000…”
2. X-linked protoporphyria cannot be excluded from the discussion for being rare, as it may be up to 10% of patients with protoporphyria. It’s certainly more common than CEP or HEP. Harderoporphyria could be considered a subtype of HCP, so that seems fine to exclude. The introduction mentions that porphyria can be from gain of function or loss of function but excludes a discussion of the only gain of function type (XLP). It should be included around lines 82, 135, and Table 2. For line 135, you could categorize EPP and XLP together as “protoporphyria” if you would like, or say that you will consider XLP a type of EPP. However, it's not okay to be excluded, as it is an important type of porphyria, even if not present in the German cohort currently.
3. For lines 101-102, I do not consider afamelanotide to be a disease-modifying therapy, so this should be edited. One group has shown a change in protoporphyrin and liver function with this, but other groups, including ours, have not. There is a disease-modifying therapy for EPP currently in a clinical trial (bitopertin), but I would not consider this clinical trial medication “available.”
Reviewer 2 Report
Comments and Suggestions for Authors
This paper describes the foundation and setup of a German registry for patients affected by nearly all forms of porphyrias. The article is soundly written: the initiative it reports is of paramount interest to all porphyrias experts, for several reasons which are properly explained in the paper. I have only a few minor issues to address:
- Footnote 1: I would appreciate to understand better why the Authors say that X-linked erythropoietic protoporphyria is described in single cases only and mostly excluded from official classifications. I would rather say that protoporphyrias, as a whole, are often considered together, FECH-related erythropoietic protoporphyria being the most common, but also including X-linked protoporphyria or even other, rarer genetic protoporphyrias (e.g. CLPX, GATA related etc.). See, for example, https://doi.org/10.3390/diagnostics12010151 or https://doi.org/10.3390/diagnostics12010151. For this reason, I don't see why the part of the registry which records EPP should not instead record all protoporphyrias, and allow for further subgroup classification.
- l. 88: please note that HFE is a single gene, so it would be more correct to say "hemochromatosis due to pathogenic HFE variants" or something alike. Even this definition is not entirely correct, e.g. because a plethora of other genes can cause hemochromatosis (HJV, TFR2, and so on), but I think it would suffice in this situation.
- l.90, "patients have an elevated risk for hepatocellular carcinoma (HCC)": has this been specifically demonstrated in EPP? aside from the expectedly increased risk due to cirrhosis.
- l.101 "modern disease modifying therapies...": I would like to know better why the Author would define afamelanotide a disease-modifying therapy for protoporhyria (if it is afamelanotide what they're hinting at). I have always thought of it as an effective treatment for symptoms, but in no way as a modifier of the natural (i.e. hepatic) course of the disease.
Furthermore, I would have appreciated a list of the laboratory values recorded, even though I trust that the registry will include all the relevant ones (e.g. homocysteinemia for acute porphyrias, iron status for both acute and erythropoietic porphyrias, and so on).
Finally, I urge the Authors not to limit themselves to German-speaking countries in Europe, when deciding whom to propose for joining the registry. As I believe that English will be the main language in which the registry is compiled, other important countries (France, Italy, Spain) may well be interested in adding their patients to a unified registry.
